

# Dynamic mercury methylation and demethylation in oligotrophic marine water

Kathleen M. Munson[1,2,3], Carl H. Lamborg[1,4], Rene M. Boiteau[1,2,5], Mak A. Saito[1]

[1]Department of Marine Chemistry and Geochemistry, Woods Hole Oceanographic Institution, Woods
Hole, MA 02543
[2]MIT/WHOI Joint Program in Chemical Oceanography
[3]Current address: Centre for Earth Observation Studies, University of Manitoba, Winnipeg, MB, R3T 2N2,
Canada
[4]Current address: Ocean Sciences Department, University of California, Santa Cruz, CA 95064
[5]Current address: Pacific Northwest National Laboratory, PO Box 999, MSIN: 58-98, Richland, WA
99352.

Correspondence to: Kathleen M. Munson (kathleenmunson@gmail.com)

**Abstract**

Monomethylmercury bioaccumulation in open ocean food webs depends on the net rate of
inorganic mercury conversion to monomethylmercury in the water column. We measured significant
methylation rates across large gradients in oxygen utilization in the oligotrophic central Pacific Ocean.
Overall, methylation rates over 24 hour incubation periods were comparable to those previously published
from Arctic and Mediterranean waters despite differences in productivity between these marine
environments. In contrast to previous studies that have attributed Hg methylation to heterotrophic bacteria,
we measured higher methylation rates in filtered water compared to unfiltered water. Furthermore, we
observed enhanced demethylation of newly produced methylated mercury in incubations of unfiltered
water relative to filtered water. The addition of station-specific bulk suspended particulate matter, a
potential source of inorganic mercury substrate, did not stimulate sustained methylation, although transient
enhancement of methylation occurred within 8 hours of addition. The addition of dissolved inorganic cobalt
also produced dramatic, if transient increases in mercury methylation. Our results suggest important roles
for non-cellular or extracellular methylation mechanisms and demethylation in determining methylated
mercury concentrations in marine oligotrophic waters. Methylation and demethylation occur dynamically
in the open ocean water column, even in regions with low accumulation of methylated mercury.



## 1 Introduction

Mercury (Hg) is a toxic metal present in small amounts in the ocean that accumulates in fish to

reach levels that pose human and environmental health risks. Because methylated Hg (collectively

abbreviated as MeHg) accumulates more efficiently than inorganic Hg, (Hg(II)), transformations into

MeHg ultimately controls burdens in upper trophic level biota. These methylated species,

monomethylmercury (MMHg) and dimethylmercury (DMHg), can be found throughout the water column,

with concentration maxima often found at depths of rapid oxygen consumption (Mason and Fitzgerald,

1993; Cossa et al., 2009; Sunderland et al; 2009; Cossa et al., 2011; Bowman et al., 2015; Bowman et al.,

2016). At these depths, production of methylated Hg has been linked to active organic matter

remineralization (Cossa et al., 2009; Sunderland et al., 2009). Water column measurements of apparent Hg

methylation rates in the Canadian Arctic Archipelago and the Mediterranean have attributed water column

methylation to bacterial processes (Lehnherr et al., 2011; Monperrus et al., 2007). However, the mechanism

of methylation in marine systems has not been determined.

Studies of obligate anaerobes from anoxic sediments have shown that Hg methylation is encoded

by a pair of prokaryotic genes, *hgcA* and *hgcB* (Parks et al., 2013), the identification of which has revealed

novel potential environments of methylation, such as sea-ice (Gionfriddo et al., 2016), and methylation

capacity across a greater diversity of bacterial species (Gilmour et al., 2013). Analysis of microbial

metagenomes from the 8 °N (150 – 800 m) and the Equatorial (50 m) stations identified sequences similar

to *hgcA* but containing base pair substitutions from known methylating bacteria sequences, the effects of

which on methylation potential are unknown (Podar et al., 2015). These findings, combined with a lack of

targeted research on *hgc* distributions in marine waters to date, demonstrate that the contribution of

bacterially mediated methylation pathways, such as *hgc*, to inventories of MeHg in oxic marine waters is

still unknown.

In order to gain insight into the mechanism of Hg methylation in the ocean, we measured

methylation rates using tracer level stable isotope additions across gradients in concentrations of

methylated Hg, total dissolved Hg (THg), dissolved oxygen, and apparent oxygen utilization (AOU)

(Munson et al., 2015) in waters from the central Pacific Ocean collected between Hawai'i (17 °N) and

American Samoa (12 °S) during the "Metzyme" cruise (Munson et al, 2015). Across this region, low



oxygen intermediate waters centered at 12 °N set apart regions of higher [MeHg] (~260 fM) in intermediate

waters (Sunderland et al., 2009) from regions of low (<100 fM) concentrations in the central and south

Pacific, with the exception of relatively elevated concentrations centered around the Equator (Munson et

al., 2015). The cruise track included regions of the North Pacific where intermediate waters incorporate

anthropogenic Hg emissions (Sunderland et al., 2009; Lamborg et al., 2014) and marine fish Hg content has

increased in recent decades (Drevnick et al., 2015).

## 2 Materials and Methods

Stock solutions of enriched isotope spikes, (MMHg: ~225 µM; Hg: ~500 uM), were prepared from

isotopically enriched HgO (Oak Ridge National Laboratory) and stored at 4°C until use.  Isotopically

enriched MM$^{198}$Hg and MM$^{199}$Hg were prepared from $^{198}$HgO and $^{199}$HgO, respectively, by methylation

with methylcobalamin (Hintelmann and Orinc, 2003). A mass of 100 µg of enriched HgO was dissolved in

10 µL HCl (conc, J. T. Baker) and diluted in 500 µL sodium acetate buffer (0.1 M, pH 5). A mass of 500 µg

of methylcobalamin (Sigma) was dissolved in 500 µL sodium acetate buffer and added to the Hg solution.

The reaction proceeded for 3 hours at room temperature before being stopped with 200 µL KBr (0.3 M KBr

in 2 M H$_2$SO$_4$). The enriched MMHg was extracted with toluene (400 µL, 3x). Extracts were combined and

dried over sodium sulfate. A 100 µL aliquot of this primary stock was dissolved in 10 mL isopropanol to

produce a secondary stock solution. Carryover of methylcobalamin as Co in the stock solutions was found

to be minimal (<1 pM) from ICPMS analysis (Thermo Element 2).

Water for incubation experiments was collected on board the *R/V Kilo Moana* between 1—24

October 2011 (Table 1). Water was collected from two depths at each station: 1) chlorophyll *a* maximum

and, 2) the oxygen minimum identified from the SeaBird CTD package data on prior water sampling casts

at each station (Table 1). Water was collected in acid-washed X-Niskin bottles deployed on a dedicated

trace metal sampling rosette using Amsteel metal-free line. Water filtration, decanting, and treatment

additions were performed under positive pressure from HEPA filter laminar flow hoods within clean

bubbles constructed from plastic sheeting. Concentrations of dissolved Hg species were measured

throughout the cruise transect (Munson et al., 2015), with the exception of 17°N, where no MMHg samples

were preserved. An estimate of MMHg at the Station 1 depths at which incubation experiments were





performed were calculated from incubation samples after correction for the added MM$^{198}$Hg spike (Table

90    1).

Incubation water was decanted under N$_2$ pressure either unfiltered or filtered through a 0.2 μm

capsule filter (47 mm, Supor polyethersulfone membrane, Pall Corporation) into 20 L acid-washed

polycarbonate mixing carboys (Nalgene). The carboys were covered in dark plastic bags during incubation

set-up to minimize exposure to light. Although hard to quantify, cells have been found in marine waters

after filtration through 0.2 μm filters in the Mediterranean and Sargasso Seas (Li and Dickie, 1985; Haller

et al., 1999) under vacuum filtration. As a result, we cannot exclude the possibility that small cells, such as

the SAR11 clade (Rappé et al., 2002), viruses, and cell material may pass through filters and therefore be

present in our filtered incubations. However, 0.2 μm filtration would potentially collect a subset of intact

cells such as SAR11 due to clogging of filters and possible decreases in actual filter pore size. In addition,

if the cells that pass through filters are sessile or, like SAR11, have slow growth rates (0.40—0.58 d$^{-1}$)

(Rappé et al., 2002), filtered waters would likely have lower cell density and less diversity in the active

metabolisms present compared to unfiltered waters.

Pre-equilibrated spikes of isotopically enriched MM$^{198}$Hg and $^{202}$Hg(II) were prepared by adding

concentrated $^{202}$Hg(II) and MM$^{198}$Hg to 0.2 μm filtered seawater and equilibrating at 16°C in the dark for

24 hours prior to use. Spikes were added using gas-tight syringes dedicated for use in isotope enrichment

experiments (Hamilton).

Incubations were performed in triplicate in 250 mL amber borosilicate glass bottles (I-Chem)

filled to the shoulder from mixing carboys leaving approximately 25 mL of headspace. As a result, *in situ*

redox conditions were not actively maintained during incubation. After water addition, pre-equilibrated

spikes of $^{202}$Hg(II) (396 fmoles) and MM$^{198}$Hg (126 fmoles) were added. Thus, spike concentrations were

added at approximately 1—14x ambient HgII and 1—34x ambient [MMHg]. Treatments of carbon (1 mM,

as succinate), inorganic cobalt (500 pM), and ½ of 2 cm punches from McLane in situ pumps (Munson et

al., 2015) were added to triplicate bottles.

Bottles for t0 measurements were fixed after isotope spike and treatment addition with 1 mL

(~0.5%, final) H$_2$SO$_4$ (conc., Fisher TM grade) and were stored at -4°C until analysis. Bottles from 17 °N,

8 °N, and 0 ° were incubated in the dark for 24 hr at temperatures maintained in refrigerators set at their



highest temperature setting (Table 1). Station 12 °S bottles were incubated in the dark up to 36 hr in a time

course study. After incubation, bottles for all time points were fixed with $H_2SO_4$ and stored at -4 °C.

Because DMHg decomposes to MMHg in acidic conditions (Black et al., 2009), all measured [MMHg]

represent the sum of DMHg and MMHg present at in each bottle at the time of acidification. Previous

methylation rate measurements have distinguished between production of MMHg and DMHg from Hg(II)

as well as the conversion of MMHg to DMHg (Lehnherr et al., 2011). Rate constants of methylation to

DMHg from Hg(II) are generally quite low between 0.03—1.22 % of those measured for MMHg

production from Hg(II) (Lehnherr et al., 2011). However, conversion of MMHg to DMHg proceeds more

rapidly and rate constants can range between 3.17—43.3% of those measured for MMHg production from

Hg(II) (Lehnherr et al., 2011). As a result, the $k_m$ values presented here represent MMHg production from

Hg(II) substrate and include any MMHg subsequently converted to DMHg.

        Samples were prepared as for MMHg determination using ascorbic-assisted direct ethylation

(Munson et al., 2014), which generates volatile methylethylmercury from the sample analyte. Analysis of

MMHg and Hg(II) concentrations and isotopic composition was performed on a ThermoFinnagan Element

2 ICPMS in the Plasma Facility at the Woods Hole Oceanographic Institution linked to a Tekran 2700

Automated Methyl Mercury Detector (Tekran, Ontario, Canada).

        Hg isotopes were measured individually and relative counts were integrated using MATLAB

scripts to quantify isotopic signals in the MMHg (as methylethylHg) and Hg(II) (as diethylHg) peaks.

Resulting methylation of $^{202}$Hg(II) and demethylation of MM$^{198}$Hg were quantified using a matrix inversion

approach to solve the system of equations (Hintelmann and Ogrinc, 2003). MM$^{199}$Hg was added as an

internal standard to samples and equilibrated for 24 hours prior to MMHg determination for isotope ratio-

ICPMS. Consistent with previous reports of methylation and demethylation rates (Monperrus et al, 2007,

Lehnherr et al, 2011), our reported rates are potential rates, which assume identical chemical behavior of

added Hg(II) and MMHg isotopes to those spcies naturally existing in the samples.

        Previous reports of methylation rates have differed in their considerations of available Hg(II)

substrate, including assuming a constant supply of Hg(II) substrate (Monperrus et al, 2007) and decreased

Hg(II) substrate availability throughout the course of incubations (Lehnherr et al, 2011). Generalized loss

of Hg(II) was not an accurate representation of Hg(II) availability in our experiments. Although not



analytically identical to total Hg measured by tin chloride reduction followed by CVAFS analysis

(Lamborg et al, 2012), we observed quantitative recovery of Hg(II) from the diethylHg peak of the Tekran

2700 instrument used for CVAFS analysis (p values <0.05 from daily standard curves). Results are

presented as percentages of methylated $^{202}$Hg(II) substrate in each incubation bottle. Apparent methylation

rate constants, $k_m$, were calculated as the percentage of $^{202}$Hg(II) substrate methylated over the incubation

time.

Potential demethylation rates were calculated assuming first order kinetics from the slope of the

linear best fit line of ln(MM$^{198}$Hg) vs. time (Lehnherr et al., 2011). Additional demethylation rates were

determined similarly from the subsequent demethylation of newly produced Me$^{202}$Hg in time course

experiments.


## 3 Results and Discussion

### 3.1 Unamended Samples

Methylation and demethylation rates were calculated from additions of species-specific enriched

Hg isotope spikes (MM$^{198}$Hg and $^{202}$HgII) that were pre-equilibrated for 24 hours with naturally occurring

ligands prior to incubation in 0.2 µm and unfiltered seawater. The dual isotope tracer technique allows us to

distinguish between the relative importance of methylation and demethylation and to explore controls on

these processes in environmental matrices (Hintelmann and Evans, 1997). Triplicate 250 mL bottles of

water were incubated in the dark for set durations of time prior to preservation. Methylation was

determined from production of Me$^{202}$Hg; demethylation was determined from loss of MM$^{198}$Hg.

First-order methylation rates were enhanced in 0.2 µm filtered water relative to unfiltered water

(Fig. 1) with the exception of low oxygen waters at the Equatorial station.  The observation of methylation

in filtered seawater was unexpected given recent focus on the role of bacterial pathways analogous to those

that occur in anoxic sediments (Monperrus et al., 2007; Lehnherr et al., 2011). However, abiotic

mechanisms of methylation have been identified from experiments conducted in distilled water and non-

marine natural waters (Yin et al., 2014; Celo et al., 2006). Because the incubation waters we collected were

influenced by the biota present prior to filtration, we do not distinguish between truly abiotic and

biologically influenced mechanisms that occurred within these waters. Furthermore, although filtration may





not remove all cells (Li and Dickie, 1985; Haller et al., 1999) we do not attribute the enhanced methylation

observed in filtered waters to cellular mechanisms. Such an assumption would require a number of

processes, such as reactivation of sessile cells, rapid rates of cell growth, and/or preferential selection of

Hg(II) methylating cells during filtration. Instead, we assume that filtration removed a majority of

heterotrophic bacteria relative to unfiltered waters.

Similar experiments in Mediterranean waters reported significant methylation in waters filtered

through 0.45 μm filters, which allows the passage of some plankton and many bacteria relative to unfiltered

waters (Monperrus et al., 2007). In contrast to our findings, which measured filtered methylation rates 138

% those of unfiltered (range 58—205 %), methylation rates in filtered Mediterranean waters, 0.8 % $d^{-1}$,

were half those in filtered waters, which averaged 1.7 % $d^{-1}$ (Monperrus et al., 2007). Thus, enhanced

methylation in filtered water may dominate in Pacific water due to its nutrient regime while larger bacteria

and perhaps eukaryotic algae may contribute in more nutrient-rich environments. Furthermore, the relative

role of non-cellular versus bacterial methylation may change in contrasting marine regions or due to

seasonal variability. Direct comparisons cannot be made until additional rates are determined in filtered

water.

In addition to enhanced methylation in filtered waters, non-cellular methylation is indicated by

rapid initial methylation of added $^{202}$Hg(II) spike within the elapsed time, between 30 min—2 hours,

between $^{202}$Hg(II) spike addition and preservation of the "t0" samples with acid (Fig. S1). This initial

methylation, like that of methylation rates over 24 hour incubations, was highest at 8 °N (subsurface

chlorophyll maximum or CMX: 8.27—12.17 % $d^{-1}$; oxygen minimum zone or OMZ: 2.45—6.84 % $d^{-1}$),

where it accounts for all methylation observed during the 24 hour incubations in the chlorophyll maximum

waters (Fig. S1). Rapid methylation has been observed in measurements of labeled Hg(II) additions to lake

water (Eckley and Hintelmann, 2006), and in previous incubations of unfiltered seawater (Lehnherr et al.,

2011). Previous marine methylation rate measurements with isotopes have exclusively used unfiltered

waters or filters that allow bacteria to pass through (Monperrus et al., 2007; Lehnherr et al., 2011), which

prevented distinction between bacterial and non-bacterial methylation in those experiments.

Methylation rates were decoupled from ambient [MeHg]. At chlorophyll maximum depths,

methylation rates were highest at 8 °N (CMX: 6.18—11.04 % $d^{-1}$) and lowest at 17 °N in the North Pacific



(CMX: 0.69 % d$^{-1}$; Fig. 1) despite nearly identical ambient [MeHg] (Table 1). At oxygen minimum depths, methylation rates generally increased as ambient [MeHg] decreased, with the exception of 17 °N in the North Pacific (OMZ: 0.74—1.10 % d$^{-1}$), where the methylation rate was lowest despite the highest [MeHg] (Table 1). As a result, methylation rates alone do not account for observed [MeHg] across gradients of

productivity.

Previous dual tracer measurements have quantified simultaneous methylation and demethylation within marine waters (Monperrus et al., 2007; Lehnherr et al., 2011), which determine the relative lifetimes of MMHg and DMHg. Our measurements of demethylation rates from loss of MM$^{198}$Hg were similar in range (0.01—0.93 % d$^{-1}$) to those measured in Arctic waters (0.23—0.59 % d$^{-1}$; 11), and higher than those

measured in the Mediterranean (0.003—0.02 % d$^{-1}$, 12). However, rapid demethylation of the added MM$^{198}$Hg spike prior to t0 was observed in some samples, which yielded non-quantitative demethylation rates in many cases (Table 2).

In waters where demethylation rates were quantified at both sampled depths, demethylation rate constants were higher in oxygen minimum waters relative to chlorophyll maximum depths (Table 2). The

enhanced demethylation in oxygen minimum waters is in contrast to the general trend of elevated [MeHg] in regions of net remineralization that is widespread in ocean basins (Mason and Fitzgerald, 1993; Cossa et al., 2009; Sunderland et al; 2009; Cossa et al., 2011; Bowman et al., 2015; Bowman et al., 2016). In addition, demethylation rate constants were enhanced in unfiltered waters (0.50 % d$^{-1}$, n = 4) relative to filtered waters (0.26 % d$^{-1}$, n = 7).

The relative importance of methylation and demethylation can be evaluated from steady-state predictions of ambient [MeHg] using measured rates to determine whether a water mass is in a state of net methylation or demethylation (Monperrus et al., 2007). The ratio of measured first-order methylation and demethylation rate constants (Table 2) predicts the steady-state ratio of MeHg to THg.. Net methylation was indicated when measured MeHg:THg exceed $k_m$:$k_d$. Net demethylation was indicated when

MeHg:THg values are lower than predicted from $k_m$:$k_d$. Net methylation has been observed during seasonal increases in primary production (Monperrus et al., 2007). Consistent with these seasonal increases we observed net methylation in chlorophyll maximum waters from the 0 ° station ($k_m$:$k_d$ = 2.5), where equatorial upwelling fuels productivity. However, net methylation was also indicated in the chlorophyll



maximum at 8 °N ($k_m$:$k_d$ = 2.9) despite its lower productivity.  All other waters were in a state of net

demethylation (average 0.3; range 0.1 – 0.9).

The relative importance of water-column methylation and demethylation was also found to be

depth dependent, with 144 ± 145 % of ambient [MeHg] predicted for by measured methylation and

demethylation rates in waters from the chlorophyll maxima in contrast to only 34 ± 39 % in oxygen

minimum waters. These results are similar to those observed in Arctic waters where deviations from

predicted MeHg:THg were attributed to limitation by supply of Hg(II) substrate for methylation (Lehnherr

et al., 2011).

The differences between depths are largely driven by differences in demethylation rates as

methylation rates are similar in waters from chlorophyll maximum and oxygen minimum depths.  Highest

[MeHg] were measured in waters from oxygen minimum depths (81—350 fM methylated Hg) relative to

chlorophyll maximum depths (26—45 fM methylated Hg; MeHg %: range 12—21% in OMZ; 7—10% in

CMX). However, these [MeHg] are not represented by the overall trends in methylation rates, which are

higher in chlorophyll maximum water than oxygen minimum waters, but do follow the trends in

demethylation when quantified (Table 2). As a result, demethylation may control [MeHg] distributions in

the marine water column.


### 3.2 Amended samples

The relative importance of methylation and demethylation on steady-state [MeHg] were further

explored by testing whether methylation rates were limited by Hg(II) substrate supply or co-factor supply

in central Pacific waters. The correlation between [MeHg] and rates of organic carbon remineralization in

North Pacific Intermediate Water (Sunderland et al., 2009) indicates that these low oxygen waters are

poised for MeHg production from Hg(II) substrate released during remineralization. The oxygen minimum

depths targeted at each station corresponded to depths of highest rates of organic matter remineralization

(Fig. S3). However, unamended methylation rates were lowest at 17 °N along the cruise transect despite

water from this station displaying the greatest drawdown of dissolved oxygen (AOU = 283 μmol kg$^{-1}$;

Table 1).





We amended incubation water with suspended particulate matter, carbon (as succinate), and cobalt additions. Suspended particulate matter was collected from oxygen minimum depths at each station using in situ pumps (Munson et al., 2015). This bulk particulate matter was tested for its ability to increase methylation by adding ½ of a 2 cm circular "punch" of the pump filter to incubation bottles.  Each punch

represented the particulate matter in approximately 70 L of seawater, a ~310–fold enrichment of particles to the incubation and 0.35 to 1.05 pmol  of added total Hg (Table S2). Succinate was added to distinguish between general stimulation of bacterial activity and the site-specific particulate matter contained in the filter punch additions.

Rather than relieving widespread Hg(II) limitation in low oxygen waters, the addition of the punch
material only stimulated methylation in filtered waters from the oxygen minimum waters at the Equator over a 24 hour incubation period (Fig. 2C). The addition of suspended particulate matter to unfiltered waters from other oxygen minimum depths did not produce clear responses and high variability was seen between replicate samples (Fig. 2). Furthermore, the addition of suspended particulate matter from McLane pump filters appeared to lower methylation rates relative to unamended samples.

Full Hg speciation measurements (Table 1; Munson et al., 2015) suggest that substantial dissolved $[Hg(II)]$ (~1 pM, from Eq. 1, Fig. S2) persist in low oxygen waters at all stations where potential methylation rates were measured.

$$[Hg(II)]_{dissolved} = [THg]_{dissolved} - [Hg(0)]_{dissolved} - [MeHg]_{dissolved} \qquad \text{(Eq. 1)}$$


The relatively high $[Hg(II)]$ imply that methylation is not solely limited by Hg(II) substrate. Instead, bulk $[Hg(II)]$ appear to provide insufficient information about Hg(II) substrate availability to mechanisms involved in methylation.

Cobalt serves as the center of methylcobalamin, the co-factor implied in *hgc*-mediated Hg
methylation (Parks et al., 2013). Inorganic cobalt (Co) is not known promote methylate mercury as a co-factor, although cobalt limitation has been observed in known methylating sulfate-reducing bacteria (Ekstrom and Morel, 2008). As a result the significant increases in methylation from additive additions of succinate and Co in oxygen minimum waters at 12 °S (Fig. 2) are surprising given they occurred in filtered



waters rather than unfiltered water. The enhanced methylation in the presence of succinate and Co in the

presence of minimal cellular material warrants further study as a potential abiotic mechanism of Hg

methylation.

### 3.3 Time course

Although 24 hour methylation rates were comparable to those measured in other marine regimes,

higher temporal resolution of sampling at the South Pacific station, 12 °S, reveals that methylation of added

$^{202}$Hg(II) is rapid, with maximum methylation reached within 6 hours in unfiltered waters (Fig. 3).

Likewise, increased methylation in to amendments with C, Co, and organic matter was rapid. Subsequent

demethylation of newly produced Me$^{202}$Hg occurs within ~12 hours of total incubation time in incubations

of unfiltered water (Fig. 3). Demethylation appeared to be a dynamic process, with demethylation of newly

produced Me$^{202}$Hg (from added $^{202}$HgII) occurring in unfiltered waters at a rate of 0.80 % d$^{-1}$ (Fig. 3).

Dynamic methylation and demethylation was observed within the first 12 hours of incubation in

both chlorophyll maximum and oxygen minimum waters (Fig. 3). The rapid transition from net methylation

to net demethylation observed at 12 °S suggests that 24 hour incubation time points are not sufficiently

resolved to document these reactions in detail at our level of spike addition in central Pacific oligotrophic

waters. This implies that our rate estimates may underestimate maximum rates of methylation and

demethylation.

Demethylation in oxygen minimum waters from 12 °S over the entire 36 hour incubation period

was influenced by the presence of particles, with newly produced Me$^{202}$Hg in unfiltered waters quickly

demethylated (0.80 % d$^{-1}$ ) after rapid production (Fig. 3b). In contrast, methylation in filtered waters is

similar in magnitude to that in unfiltered waters but persists over the course of the incubation (Fig. 3). The

use of dual tracers in this time course experiment reveals that demethylation controls [MeHg] as the

incubation approaches steady-state, indicating an important role of demethylation in determining [MeHg]

in the marine water column and its availability for bioaccumulation.

**3.4 Mercury Methylation and Organic Matter Remineralization**



Although elevated ambient [MeHg] have long been recognized as existing in regions of high oxygen utilization (Mason and Fitzgerald, 1993; Cossa et al., 2009; Sunderland et al; 2009; Cossa et al., 2011; Bowman et al., 2015; Bowman et al., 2016), gaps persist in our understanding of how organic matter remineralization influences methylation. While low [MeHg] in surface and deep waters are thought to be

the result of active demethylation by both photodemethylation and biological demethylation (Cossa et al., 2009; Sunderland et al., 2009; Monperrus et al., 2007; Lehnherr et al., 2011) subsurface peaks in [MeHg] are thought to be controlled by two factors: 1) the release of Hg(II) substrate from sinking organic matter remineralization (Cossa et al., 2009; Sunderland et al., 2009), and 2) enhancement of microbial loop activity (Heimbürger et al., 2010). Microbial loop activity has been suggested to dominate subsurface

MeHg dynamics especially in oligotrophic waters (Heimbürger et al., 2010).

The subtropical gyre waters at 17 °N and 8 °N were oligotrophic regions exhibiting nitrogen stress while 12 °S was within a region exhibiting both low nitrate concentrations and iron stress (Saito et al., 2014). Total concentrations of THg in the North Pacific reveal a sharp transition between North Pacific Intermediate Water and intermediate waters south of 17 °N (Sunderland et al., 2009; Munson et al., 2015).

Within the Equatorial Pacific, we observed an area of low Hg species concentrations, and significant reduction of Hg(II) centered at 12 °N and extending toward the 8 °N incubation station that corresponds to regions of denitrification between 200—400 m, as indicated by calculated N* values (Munson et al, 2015). The reduction of Hg(II) appears to limit the supply of Hg(II) to depths below 400 m, consistent with conceptual models of methylation due to substrate supply (Sunderland et al., 2009; Lehnherr et al., 2011).

However, the addition of suspended particulate matter from McLane pump filters to bottle incubation did not enhance potential methylation in waters collected from this site. These observations suggest that delivery of Hg(II) by bulk organic matter does not limit MeHg production in North Pacific waters. This observation is consistent with some field data, where MeHg:Hg below the photic zone is not elevated compared to other regions of the water column (Sunderland et al., 2009; Hammerschmidt and

Bowman, 2012), including regions adjacent to our incubation stations (Munson et al., 2015). In contrast, the addition of Co and additive additions of Co and succinate were able to promote methylation, most notably in South Pacific waters, which suggests a more specific control, such Hg ligand binding or cofactor availability, on methylation rather than bulk Hg(II) supply. Therefore, the dominance and rapidity of initial



methylation in our incubations may indicate the importance of ligand exchange with Hg(II) substrate in

oligotrophic waters.

In addition to the potential role of organic matter in controlling Hg methylation, our results

indicate an important role of particulate matter in demethylation and thus water column [MeHg]. The

enriched Hg species spikes used as tracers were pre-equilibrated in filtered seawater for 24 hours prior to

addition to bottle incubations in order to allow Hg to bind with DOM complexes in an attempt to more

closely mimic seawater speciation of Hg (Lamborg et al., 2004). Given that the MM$^{198}$Hg enriched stock

solution was stable over the 24 hour pre-equilibration with filtered seawater, the rapid demethylation

observed upon addition of the spike to seawater at tracer levels in incubations was surprising. As the

filtered seawater used for pre-equilibration differed from incubation water only in its ratio of enriched Hg

to ligands, the rapid demethylation may further indicate the role of ligands in controlling Hg species

dynamics.

**4 Conclusions**

We observed active Hg methylation in filtered waters, which suggests a potential role for non-

cellular methylation in oligotrophic marine systems. Measured Hg methylation dynamics in the central

Pacific Ocean were greater in filtered water, implicating non-cellular or extracellular processes as the

primary mechanisms of mercury transformation in the marine water column. Determining the extent of this

non-cellular methylation relative to bacterial methylation is important for understanding the potential for

MMHg entry into the marine food web.

The methylation potential encoded by *hgc* genes provides an important tool for determining the

extent of biotic methylation. Failure to identify functional *hgc* sequences from Metzyme cruise stations

does not exclude the possibility that these genes are present in such low abundance as to escape detection

nor the ability of the observed *hgc*-like sequences (Podar et al., 2015) to encode methylation. Furthermore,

although these genes are absent from current Tropical Pacific metagenomes, functional *hgcA* and *hgcB*

genes have been identified in other marine regions that are influenced by continental shelf sediments,

permafrost, and estuarine waters (Podar et al., 2015). In these regions, direct biotic transformations of

mercury may be more important compared to the oligotrophic Pacific.




While cellular methylation does not appear to dominate oligotrophic waters in the Central Pacific, our incubations suggest a primary role of Hg-ligand binding in controlling steady-state [MeHg]. We identified demethylation as the primary control on [MeHg] across wide gradients in productivity across a

section of the Pacific. However, the lifetime of MeHg in seawater appears to be influenced by the presence of organic matter, including the relative concentration of organic matter to MeHg.

Author contributions: KMM, CHL, and MAS designed research; KMM, CHL, and RMB performed research; KMM and CHL analyzed data; KMM and CHL prepared the manuscript with contributions from

RMB and MAS.

The authors declare that they have no conflict of interest.

**Acknowledgements**

This work was funded by the National Science Foundation in a Chemical Oceanography Program Grant (OCE-1031271) awarded to CH Lamborg and MA Saito and a Graduate Student Fellowship to KM Munson. We thank Vic Polidoro, Trevor Young, Captain Drewry, and the crew of the *R/V Kilo Moana*.

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





**Figures:**

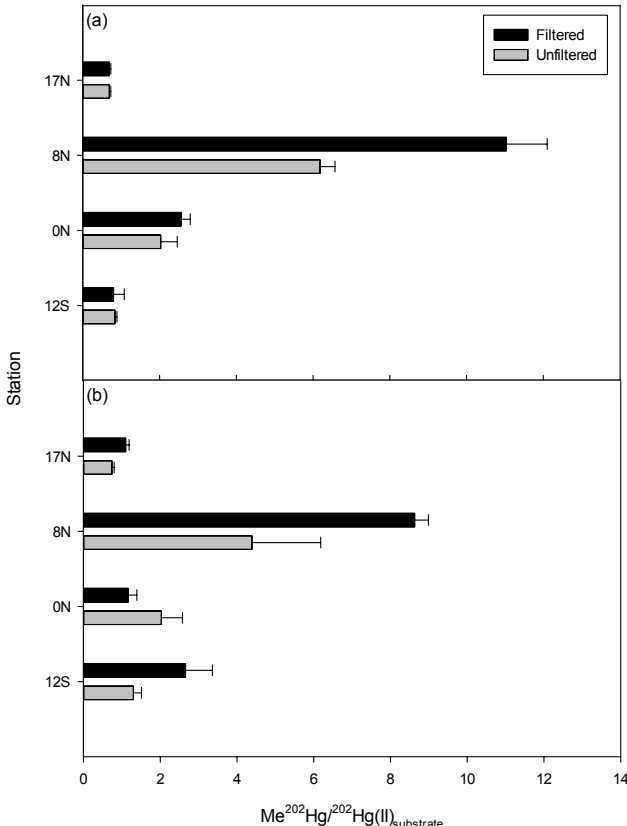

Figure 1: Methylation of Hg(II) in Tropical Pacific waters from triplicate bottle incubation (± 1 SD), as the percentage of Me$^{202}$Hg produced from excess $^{202}$Hg(II) substrate from isotope spike additions, for Metzyme cruise stations from the North to South Pacific. Methylation was measured in 0.2 µm filtered and unfiltered from the chlorophyll maximum (a) and oxygen minimum (b) depths at each station.





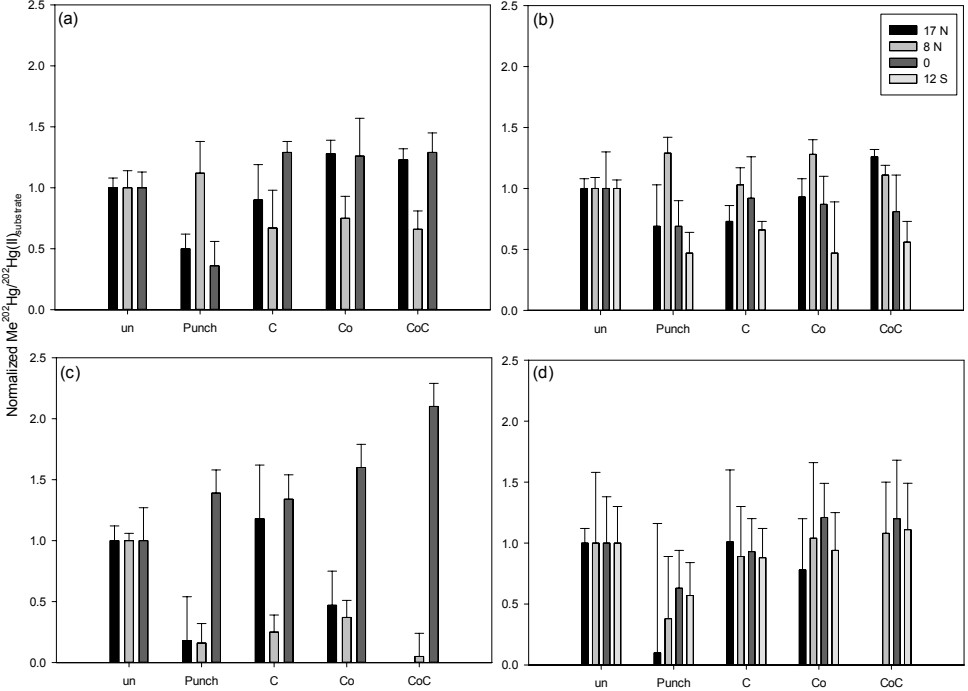

Figure 2: Normalized methylation rates from 24 hour incubations. Methylation rates from triplicate
incubation bottles (±1 SD) normalized to unamended (un) water from the chlorophyll maxima (a, b) and
oxygen minima (c, d). Treatments are abbreviated as follows: cobalt (Co), succinate (c), McLane pump
sections (punch). The error bars represent one standard deviation from triplicate incubation bottles.





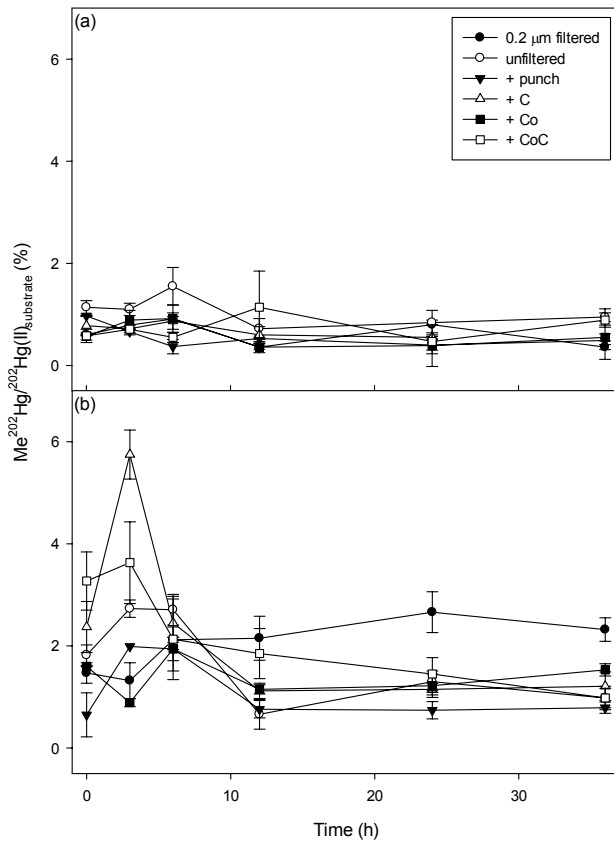

Figure 3: Methylated mercury production over 36 hours at 12 °S. Methylation rates from triplicate
incubation bottles (±1 SD) in water from the chlorophyll maximum (a) and oxygen minimum (c) depths in
the South Pacific. Newly produced MeHg only persisted beyond 12 hours in 0.2 μm filtered water (B).




| Sta | Depth (m) | | Temp (°C) | $Temp_{inc}$ (°C) | $O_{2diss}$ (μmol/kg) | $AOU^a$ (μmol/kg) | THg (pM) | MeHg (fM) |
|---|---|---|---|---|---|---|---|---|
| 17 °N | CMX | 120 :150[b] | 22 | 19-26 | 214:205 | - | 1:0.59 | 41 |
| | OMZ | 500 | 7 | 13 | 20 | 282.9 | 1.66 | 349 |
| 8 °N | CMX | 75-80 | 21 | 23 | 218 | - | 0.32 | 31 |
| | OMZ | 200 | 11 | 13 | 15 | 264.1 | 1.4 | 152 |
| 0 ° | CMX | 50 | 26 | 23 | 202 | - | 0.18 | 36 |
| | OMZ | 500 | 8 | 14 | 37 | 252 | 1.1 | 255 |
| 12 °S | CMX | 60 | 28 | 13-16 | 206 | - | 0.22 | 15 |
| | OMZ | 175 | 24 | 13-16 | 162 | 51.7 | 0.35 | 23 |
| | OMZ | 400 | 11 | 13-16 | 116 | 163.1 | 0.67 | 81 |

Table 1: Water column characteristics for Pacific Ocean waters from which potential mercury methylation rates were measured.

[a]AOU is a poor proxy for oxygen utilization in waters above 100 m due to gas exchange and photosynthesis. Values are therefore omitted for these depths.

[b]Water was combined from 120 m and 150 m (1:3 mixture) due to water budget limitations.

[c]MMHg concentrations were calculated from ambient $^{200}$Hg in t0 incubation bottles after subtraction of contribution from added MMHg spike.





| Sta | Depth (m) | $k_m$-0.2 µm filt ($\times 10^{-2}$ d$^{-1}$) (± 1 SD) | $k_m$-unfilt ($\times 10^{-2}$ d$^{-1}$) (± 1 SD) | $k_d$-0.2 µm filt (d$^{-1}$) | $k_d$-unfilt (d$^{-1}$) |
|---|---|---|---|---|---|
| 17 °N | CMX | 0.43 ± 0.11 | 0.43 ± 0.02 | 0.07 | nd |
| | OMZ | 0.95 ± 0.08 | 0.64 ± 0.05 | 0.07 | nd |
| 8 °N | CMX | 10.19 ± 0.98 | 5.71 ± 0.36 | 0.04 | 0.62 |
| | OMZ | 7.40 ± 0.31 | 3.77 ± 1.54 | nd | 0.93 |
| 0 ° | CMX | 2.19 ± 0.20 | 1.74 ± 0.37 | nd | 0.07 |
| | OMZ | 1.00 ± 0.19 | 1.74 ± 0.47 | 0.54 | nd |
| 12 °S | CMX | 0.69 ± 0.28 | 0.72 ± 0.02 | 0.22 | nd |
| | OMZ | 0.20 ± 0.04 | 0.49 ± 0.08 | 0.20 | nd |
| | OMZ | 2.28 ± 0.70 | 1.12 ± 0.21 | 0.68 | 0.39 |

Table 2: Potential methylation ($k_m$) and demethylation ($k_d$) rate constants from triplicate bottle incubations of Tropical Pacific waters in filtered and unfiltered water.