# Peer review of "Dynamic mercury methylation and demethylation in"

_Biogeosciences, 2018_

## Referee Comment (RC1) · Anonymous Referee #1 · 14 May 2018

Manuscript "Dynamic mercury methylation and demethylation in oligotrophic marine water" by Munson et al. presents information that the mercury research community has been waiting for. This study addresses Hg transformations that occur in seawater. Specifically, this study is addressing these transformations in nutrient depleted waters in Equatorial Pacific Ocean and the research took place during Metzyme cruise in 2011. The incubation experiments used seawater collected from depths that were autotrophic organisms were most abundant i.e. chlorophyll maximum, and from depths where oxygen was most rapidly consumed mostly due to intensified organic matter remineralization. I command author's choice of their experimental plan as autotrophic as well as heterotroph driven processes have been hypothesized as strongly coupled with MeHg in the ocean. Other elements of Hg cycling have been also shown to be

coupled with biological processes. Analytical work has been designed carefully. Most important details are included and the readers should be able to imagine how the study was conducted and how analyses were conducted with some exceptions, which I will list in my specific comments. Munson et al. have conducted incubation by triplicating bottle, this has not been done that way and so it is great to be able to see the error bars on data representing specific time points. One can notice that error bars are not non-negligible further reinforcing the importance of replication in bottle incubations – I command Munson et al. for following such procedure. To understand differences between formation of MeHg (Note: as authors have done, I am combining into this category any organic Hg form that would be detected as monomethylmercury or MMHg) and degradation of MMHg Munson et al. have computed potential reaction rates as done in some previous studies. Authors refer to the Lehnherr et al. articles in reference to their calculations of kd, the demethylation rate however surprising little detail is provided for a reader to fully understand how the rates presented in the Munson et al. study have been calculated. This description needs to be improved and authors need to present which time points were included in rate calculation. It is becoming even more complicated for the calculated for kd as authors seem to use two approaches to calculate them i.e. 1). based on a spike of isotopically labeled MM198Hg and 2). based on degradation of "newly formed" [from a spike of 202Hg(II)] MMe202Hg during the time course incubation experiment. I find that dedicating more attention to proper description of this component is critical as all the Hg related results depend on these calculations. One other note I want to make in relation to the reporting of the "potential methylation rates" is that authors well realize that the these estimated rates are very likely inaccurate. The uncertainties associated with the reported rates are merely due to differences in detected MMHg between replicate bottles, while other factors might influence the results too, they are just not considered here. At this point there is such limited body of work on this subject and my feeling is that methods to accurately and confidently provide rates of Hg transformations are not fully worked out – this work is not straightforward and as authors mention the other publications reported on "rates"

that were determined based on somewhat different procedures. I would urge the authors to caution readers, especially those from the modeling community, to consider that even though called "potential rates" might be misrepresenting the actual rates. More research using water incubations with Hg stable isotopes must be published and the community needs to work out "common practices" in this type of research. Still, all this caution and hesitation to take the "rates" values seriously, I believe that the data set generated in this study is impressive, novel and deserves publication. Surely, this study pushes the field forward. However, there are some items that must be addressed before this manuscript turns into a publication. This study provides time series with good replication – you could do statistics to compare curves – why wasn't this done? I don't have many comments for the "Introduction", I think that authors have provided a solid background. However while authors notify the reader in the last paragraph of this section, they were aiming "to gain insight into the mechanisms of Hg methylation in the ocean". Sure they have provided some insight. My criticism is that the study isn't presenting clear hypotheses that authors were testing for. It is clear that hypotheses were posed but authors have forgotten to include them into this manuscript. I am a strong believer in hypothesis driven research, especially in experimental research type. Statement of goal and hypotheses would help structure the discussion of results which is at the moment following the different kinds of experiments there went on during that cruise – 3.1 Unamended samples, 3.2 Amended samples etc. – these titles are not very exciting. I already know about all these experiments because I read through the materials and methods – I would advise to think of a title that is more informative i.e. provides a clue about the main message/finding/phenomenon that experimentation provided supporting evidence for. I don't see why this article has to be split up in the current fashion. Again, if you had hypotheses then these could be used for a structure of discussion and the flow would be largely improved. I would recommend that authors consider changing. It is a great study, my recommendation is to help making it even greater (of course that is in my opinion)!

Beneath, I will list my more specific comments: Materials and methods Line 86 – should

it be "Hg species" or "Hg form"? - consider throughout the manuscript. Line 112 –" $\frac{1}{2}$ " would be nicer if spelled out, I think. (- > this is also in line 259) Line 114 – Instead just "T0" perhaps it would read nicer if an additional description appear near e.g. T0 = time of the initial sampling or something like that. Line 116 – How about "Equator" instead of "0°" ? Line 117 – I am not sure why the incubation was set at the highest temperature setting? Please explain in text. Line 118 – Provide final concentration of sulfuric acid in seawater. Line 123 – Apropos "generally low" – This give a feeling as if kinetics of DMHg degradation and even MMHg degradation in different seawaters were well established but the first one was measure in only one study thus far! Please revise this text. Lines 126-127 – Make it clear that this is an assumption – reword appropriately. Line 131 - Provide more information about your Matlab script - is it freely available? Who has written the code etc. Line 136 – present your assumption about the amount of MM198Hg spike used as an internal standard. Line 139 - You highlight the "potential rates" but you have also assumed the linear reaction, which is a far cry from what we see in your data. Again, clarity to how the rates were determined would be beneficial. Line 146 – Do you mean that the recoveries were comparable? – This sentence is not clear to me. Line 148 – Do you decide to use the term "apparent" or "potential"? Decide to choose one. I actually think that "apparent" is a more appropriate. You would need to check throughout the manuscript. I think these rates are apparent based on specific experimental design and should be treated with a grain of salt.

Line 151-154 - This paragraph needs to be improved and more information is required as I already noted in my general comments.

Results and Discussion

First paragraph reads like "material and methods"

Line 162 – Yay for triplicates!

Line 165 - What exactly is the 1st order here? Fig. 1 doesn't show rates. Again, maybe I am unsure because you have not really specified how the potential rates were

calculated. If the publisher does not limit figures then why not include all the data – show results from all incubations?

Line 175 - You mean reactivation of dormant cells? Sessile doesn't seem right in this context.

Line 177 - This sentence feel unfinished. Could you put a coma there and finish it off with something similar to this "therefore decreasing the likelihood of microbe-mediated Hg methylation".

Lines 182- 183 - You say: "…enhanced methylation in filtered water may dominate in Pacific water" but this is too general. Please be sure that this statement is only relevant to your study region.

Line 223 – Remove extra period.

Lines 237-244 - This paragraph is not coherent. The topic sentence is talking about the importance of demethylation but this is not developed any further. In the topic sentence specify differences in what.

Line 253 - If Fig. S3 is so important and it seems that authors use it to support their claims then why not include it into the main text? Also, this figure could use O2 information interpolated on top of the OCRR values.

Line 258 – in situ should be in italics

Line 266 - it should be "filtered particulate matter" as it was no longer in suspension.

Line 271 - This comment is for the caption in Fig. S2 – please change wording. For example: Concentration of Hg(II) as calculated by balancing measured dissolved Hg forms i.e. THg, MeHg, Hg(0) based on equation 1 in the main text.

Equation 1 – where is the MeHg diss. From? Why isn't it presented here? Also, where are the other measured values from?

For the paragraph beginning at Line 270 - I think that the discussion here is poor and should be expanded. There have been studies discussing the issue of availability whether bioavailability or chemical availability/reactivity.

Line 280 – Wording here is awkward – please revise

Line 282 – Again, please revise wording "additive additions" ??? That doesn't sound right.

Line 292 - Isn't C succinate? If so I would just provide that name and remind the reader that it was a generic source of carbon.

Line 296 - What exactly do you mean by "dynamic methylation" – it sounds scientific but it delivers no meaning. Please consider changing it. You can simply describe the pattern of how and when things changed, just the way you did it in the second sentence.

Line 317 – But the release of Hg(II) during the two processes i.e. sinking and reminer- alization are connected because enhanced microbial processes are associated with sinking particles that when nearer to surface are more organic matter rich.

Lines 330 - till conclusions – great discussion on the shortcomings of spiking. Ligands need to be addressed in future research.

Line – 360 – this is the first I read about any effort to identify genes from Metzyme- this comes out of nowhere. The whole issue of genes here is completely unexpected – I don't see how this fits as a conclusion to this particular paper. I recommend rethinking the conclusion.

Fig. 3 – I would get rid of all the lines – they blur the figure, which already contains a lot of symbols. Perhaps you can consider splitting these two panels into more small pan- els? - it would show patterns more easily and then you can keep lines connecting data for specific time points. Scale on y-axis in a) is too large. I would increase resolution of the x-axis.

---

## Referee Comment (RC2) · Anonymous Referee #2 · 16 May 2018

This paper studies mercury methylation and methylmercury demethylation in ocean water using isotope enriched mercury species. Using a variety of experimental approaches, the authors determine the magnitude of species transformation, including time series and amending water with other substrates such as organic carbon and cobalt. The use of isotope enriched species, combined with ICP/MS quantification allows the monitoring of both processes simultaneously in the same sample. The method is also sufficiently sensitive to work with spikes that are close to natural levels of mercury, providing as much environmental relevance as possible. I especially like the novel comparison of filtered with unfiltered waters, which required a meticulous experimental work to avoid contamination during handling of samples. Nevertheless, this study is extremely ambitious, and goes to the limit of what appears to be experimentally possible

today. One challenges of this type of study is the uncertainty regarding actual conversion rates that can be expected in previously untested environments, making the experimental design a bit of a guess work. The authors chose to incubate the majority of treatments for 24 hours, presumably to ensure detectable transformation results. However, as their time series suggest, it appears that these waters already showed measurable methylation and demethylation within 3-6 hours, after which the initially formed MMHg was again demethylated. Hence, I am concerned that the obtained data over the 24 hour incubation period cannot be considered "rates" of methylation or demethylation. In my opinion, the 24 hour data rather represent net MMHg formation. The time series suggest that there are multiple "rates" that are in play within 24 hours. An initial fast methylation, followed by a loss or at least a rapid decline in methylation activity. Instead, the demethylation process became more active at this point. I would interpret the data as if there was rapid initial methylation, but that the methylating agents or processes were exhausted with a few hours and the system never got into a (new) steady state. If this interpretation were correct, the calculation of rate constants is not possible using the data as suggested in the manuscript and the interpretation of [MMH]/[Hg] to Km/Kd ratios becomes questionable, though I agree that the results look intriguing, but maybe the similarity is generated for a different reason.

For clarification: it is not clear, how the 24 hour conversion rates were calculated. Is this obtained from the difference in MMHg concentrations between t(0) and t(24) or rather the difference between the nominal spike concentration and the concentration at t(24). It would be helpful, if the authors can provide the raw data in tabulated form in the appendix. From figure 3 it almost appears as if there frequently is no net change in concentration between t(0) and t(24), if this is the case, how can you calculate a methylation rate?

Specific comments: Title: What is supposed to be conveyed by adding "Dynamic" to the title? Is there also a "lethargic" methylation? Or are you referring to the "Dynamics" of mercury methylation . . .?

L 76: terminology: rather than referring to "enriched isotope spikes" use "isotope enriched spikes. It is the spike that is enriched with isotopes not the other way around.

L89: how exactly did you calculate ambient MMHg concentrations? What is meant by "correction for the added MM198Hg spike"?

L112: the concept of the "punches from McLane in situ pumps" requires more explanation. This appears to be lab lingo, which his incomprehensible to me at this point, though later on the authors shed a bit more light of what this likely means.

L116: given that incubations were not performed at in-situ T, I am missing a discussion how this might have affected the outcome, since T changes alone could alter bacterial activity, leading to changes in steady state MMHg levels.

L145: it is an interesting concept to determine Hg(II) through direct ethylation. However, to be convinced that this is actually a viable method, I would require more QA/QC data, especially ethylation blanks. I would assume that reagents used in the methods carry some inorganic Hg background (buffers, acids, the ethylation reagent . . .)

L150: How are you determining a first order decay constant from at best two data points? I assume that the two points (t(0) and t(24)) themselves carry considerable uncertainty. Given than the exponential relationship, this should translate in rather large uncertainty of the resulting linear relationship and rate constant. Even if this calculation was doable (which I somehow doubt), at the very least, you should provide an uncertainty estimate, which his suspiciously absent for Kd values of table 2, while on the other hand uncertainties for Km are provided. Equally concerning, I can't find a single data point (in a table or on a graph) for measured MM198Hg levels before, during or after incubation. This needs to be provided in order to ascertain the conclusions drawn in this paper.

L158: as mentioned earlier, I think it is misleading to claim that this experiment determined "rates". Instead, it determined the net methylation that occurred over a 24 hour

incubation period. In the absence of a time series showing a continuous change in concentration over time, I like challenge the idea that this dataset allows the calculation of rates, let alone rate constants. Here, the "rate" is obtained by drawing a straight line between two arbitrary points on the time axis. If the authors had chosen to incubate all samples for 6 hours, we would be facing very different "rates".

L210: are you sure that MMHg was indeed demethylated prior to t(0)? Have you considered other loss mechanisms, e.g. adsorption to container walls? Did you try to determine the T198Hg concentration in these samples? If there was demethylation, leading to 198Hg(II), it should show up during a total Hg determination or in the diethylHg peak of the chromatogram. If absent, what does this say about the demethylation mechanism? Would that mean the product of the demethylation is 198Hg(0)? Is that possible? Where did the 198Hg isotopes go, if they are no longer detectable as MM198Hg?

L218. Be careful to not confuse "rates" with "rate constants". Demethylation rates may be expressed as the % loss per day, but this is a rather unusual expression for a rate constant, which for first order processes, has the unit of d-1 (per day). Why do you add "%" at this point?

L223: I agree with this concept, but I disagree in that the data obtained here are indeed "rate constants" instead, they are more net conversions over 24 h of incubation.

L243: given the absence of any actual data on MM198Hg concentrations, it is difficult to validate this conclusion.

L259: this description of the "punch" should go to the methods section.

L285; this is an intriguing observation. I'd be curious if this an experimental artifact and artificial or if this indeed points to environmental relevance for the methylation process. Certainly worth exploring in more detail.

L294+296: "appears to be a dynamic process" what is "dynamic" in this process?

Seems to be an unnecessary filler. Please, omit "dynamic".

L295: how did you calculate the rate of MM202Hg demethylation? Please, explain.

L297: I completely agree that the 24 hour incubations don't offer the resolution which would allow rate estimates with any certainty. This is not a critique of the experiment, but merely an observation. As mentioned earlier, these experiments require an educated guess about appropriate incubation periods and one only discovers after the fact, how good the initial guess really was. But rather than risking an overinterpretation of the data, the authors should rephrase their conclusions accordingly. Take into account the inherent limitations of this type of study.

L343: this observation is indeed puzzling. Can it have something to do with the acidification that is used to stop incubations?

L352: another unnecessary filler: "active" seems the wrong word here, unless there is also a "passive" methylation process.

L369-366: where is this discussion coming from? I fail to connect the body of this research to hgc genes.

L367: how do you know that cellular methylation is not important, when you only determined net 24 hour methylation, rather than studying what is going on in the first 6 hours, were cellular processes may very well be important. But after 6 hours cells die (for whatever reason) and only appear to be unimportant (in the artificial setting of a closed 250 mL bottle).

Figure 2: What is the difference between panel a+b and c+d? There is no legend for panels a+c. Was the concentration of the Hg(II) substrate determined (how?) or is this the nominal spike concentration?

Figure 3: the chosen presentation makes it very difficult, if not impossible for most treatments to decide if concentrations after 24 hours are smaller or larger compared to the t(0) starting point

Table 1: typo for THg of 17N CMX: 1:0.

Table2: Why are there no uncertainty estimates for Kd values?

---

## Author Comment (AC1) · 23 Jul 2018

Author comment #1

Response to general comments: We authors agree with the Referee #1's comments that the calculated potential methylation and demethylation rates over a 24-hour period fail to encompass the complexity of the changes occurring in the experiment. Our replicates and higher temporal resolution at the 12 °S station presented in the manuscript clearly revealed that rate measurements calculated from a single bottle and/or time point can mask variability that are important to recognize as isotopic tracers continue to be used to estimate Hg methylation and demethylation in further studies. We believe the variability seen in our experiments were important to recognize before comparisons

can be made between water bodies.

Both referees rightly pointed out that we provided too few details concerning our "rate" calculations, especially demethylation. This was a significant oversight in our initial submission and has been addressed in the subsequent revision in the following ways: -details of the calculations are included in the materials and methods sections, including differentiation between calculations used to distinguish between demethylation of 198MeHg and 202MeHg. -in response to comments from Referee #2, we have focused our data presentation and discussion on the % loss of MeHg rather than on calculated 24 hour demethylation rates since these often fail to describe our observations.

In addition, we have adjusted the introduction to clearly identify the following hypotheses that guided our experimental design: 1. If intracellular bacterial methylation were the primary source of MMHg to marine waters, we expected higher methylation rates in incubations of unfiltered water compared to filtered water from the equatorial Pacific. 2. If in situ methylation is the primary source of MMHg to the marine water column, we expect elevated rates of net methylation in oxygen minimum waters relative to chlorophyll maximum waters due to the relative concentrations between these depths. 3. If sinking particles provide Hg(II) substrate to oxygen minimum depths, we expect enhanced methylation in waters amended with additional particulate material.

Line 86 – should it be "Hg species" or "Hg form"? - consider throughout the manuscript. Total Hg, Hg(0), MMHg, and DMHg were analyzed as chemical species of mercury in seawater samples. Thus "species" is the correct terminology. I am not aware of the exact definition of a chemical "form."

Line 112 –" 1/2 " would be nicer if spelled out, I think. (- > this is also in line 259) Changed.

Line 114 – Instead just "T0" perhaps it would read nicer if an additional description appear near e.g. T0 = time of the initial sampling or something like that. Changed to "timepoint of incubation onset (t0)."

Line 116 – How about "Equator" instead of "0_" ? Changed.

Line 117 – I am not sure why the incubation was set at the highest temperature setting? Please explain in text. Changed to: "Bottles from Stations 17° N, 8° N, and the Equator (0 °) were incubated in the dark for 24-hr in refrigerators maintained at temperatures closely matching the in situ temperatures from which the samples were taken (Table 1)."

Line 118 – Provide final concentration of sulfuric acid in seawater. Changed to "H2SO4 (0.5 % final volume)."

Line 123 – Apropos "generally low" – This give a feeling as if kinetics of DMHg degradation and even MMHg degradation in different seawaters were well established but the first one was measure in only one study thus far! Please revise this text. This is an important point. We have changed the text to reflect the fact that a single study has distinguished between Hg(II) to DMHg methylation, Hg(II) to MMHg methylation, and MMHg to DMHg methylation. We want to emphasize the point that our measured methylation or demethylation cannot be directly compared to the production or demethylation of MMHg and DMHg, if analyzed separately.

Lines 126-127 – Make it clear that this is an assumption – reword appropriately. With the changes to the paragraph in response to the previous line comments, we adjusted the paragraph to reflect that our measurements of Hg(II) and MeHg encompass any conversions between Hg(II), MMHg, and DMHg.

Line 131 - Provide more information about your Matlab script - is it freely available? Who has written the code etc. We wrote the MATLAB scripts and have included them in the supplemental section with an example calculation.

Line 136 – present your assumption about the amount of MM198Hg spike used as an internal standard. The internal standard used was MM199Hg, not MM198Hg. As a result, there is no assumption about the amount used for the internal standard. Instead,

the amount of MM199Hg added (25 fmol) was used to monitor the MeHg ethylation recovery.

Line 139 - You highlight the "potential rates" but you have also assumed the linear reaction, which is a far cry from what we see in your data. Again, clarity to how the rates were determined would be beneficial. Valid point. As pointed out by Referee #2 as well, we relied far too heavily on our 24 hour "rates" and, as discussed, provided too few details concerning our exact calculations. We have adjusted our "rates" to a more appropriate use of methylation and demethylation along the time course of our incubations (as % methylated and % demethylated) and our description of "rates" accordingly.

Line 146 – Do you mean that the recoveries were comparable? – This sentence is not clear to me. We used 202Hg(II) determined as diethylHg from the chromatograms of the Tekran 2700-ICPMS system to estimate the 202Hg(II) substrate in each incubation bottle. We observed quantitative recovery of Hg(II) from standard curves of THg analyzed using the Tekran 2700.

Line 148 – Do you decide to use the term "apparent" or "potential"? Decide to choose one. I actually think that "apparent" is a more appropriate. You would need to check throughout the manuscript. I think these rates are apparent based on specific experimental design and should be treated with a grain of salt. As mentioned in our response to general comments, we have adjusted our use of rates to clearly distinguish between our 24 hour rates and the much faster time scales of methylation and demethylaton observed or indicated in our experiments. In our initial submission, we generally used "potential" to refer to the fact that any isotope tracer studies assume identical availability of the added tracer Hg(II) or MMHg with that of in situ Hg(II) and MMHg. We used "apparent" to reference our measured rates are not "potential" since the 24 hour incubation time is inappropriate for our study. Our revisions now reduce our interpretation of the 24 hour rates.

Line 151-154 - This paragraph needs to be improved and more information is required as I already noted in my general comments. Agreed, as discussed in our response to your general comments.

Results and Discussion First paragraph reads like "material and methods" Agreed and have removed this section as it is repetitive.

Line 162 – Yay for triplicates! Agreed.

Line 165 - What exactly is the 1st order here? Fig. 1 doesn't show rates. Again, maybe I am unsure because you have not really specified how the potential rates were calculated. If the publisher does not limit figures then why not include all the data – show results from all incubations? The first order approximation indicates the dependence on concentrations of MMHg and Hg(II) for demethylation and methylation. Studies to date have used the calculation of first order rate constants

Line 175 - You mean reactivation of dormant cells? Sessile doesn't seem right in this context. Changed to "dormant."

Line 177 - This sentence feel unfinished. Could you put a comma there and finish it off with something similar to this "therefore decreasing the likelihood of microbe-mediated Hg methylation". We have added the recommended terminology.

Lines 182- 183 - You say: ": : :enhanced methylation in filtered water may dominate in Pacific water" but this is too general. Please be sure that this statement is only relevant to your study region. While our findings are not general to the Pacific, our stations encompassed regions that are more oligotrophic than other regions where rates have been measured. We have revised to be more specific to our oligotrophic stations.

Line 223 – Remove extra period. Removed.

Lines 237-244 - This paragraph is not coherent. The topic sentence is talking about the importance of demethylation but this is not developed any further. In the topic sentence specify differences in what. We adjusted the paragraph to reflect our evaluation

of how relative rates of methylation and demethylation contribute to measured MeHg concentrations in seawater. Our aim is to emphasize the fact that MeHg concentrations in seawater are currently attributed to differences in methylation. However, our findings suggest that demethylation is rapid and the specific controls on demethylation warrant further study.

Line 253 - If Fig. S3 is so important and it seems that authors use it to support their claims then why not include it into the main text? Also, this figure could use O2 information interpolated on top of the OCRR values. The OCRR data were calculated and presented in our previous manuscript (Munson et al, 2015) focused on the measured Hg species in this region, thus we feel that a reprint of those data are not needed for the main text.

Line 258 – in situ should be in italics Changed.

Line 266 - it should be "filtered particulate matter" as it was no longer in suspension. Changed throughout the manuscript.

Line 271 - This comment is for the caption in Fig. S2 – please change wording. For example: Concentration of Hg(II) as calculated by balancing measured dissolved Hg forms i.e. THg, MeHg, Hg(0) based on equation 1 in the main text. Changed to "Concentrations of Hg(II) as calculating from full dissolved Hg speciation measurements (THg, Hg(0), MMHg, and DMHg) throughout the cruise transect based on Equation 1 in the main text."

Equation 1 – where is the MeHg diss. From? Why isn't it presented here? Also, where are the other measured values from? Total Hg, Hg(0), MMHg, and DMHg were measured and presented in our 2015 manuscript as cited in the current manuscript. Those data are used here to calculate Hg(II), the assumed primary substrate for methylation, here using Equation 1 as presented. We adjusted the description to make that more clear.

For the paragraph beginning at Line 270 - I think that the discussion here is poor and should be expanded. There have been studies discussing the issue of availability whether bioavailability or chemical availability/reactivity. As noted in our response to the general comments, we have adjusted our revisions to address our guiding hypotheses for this work As a result, we have adjusted our discussion of bioavailability, primarily citing the results of Schaefer and Morel, 2009, which informed the design of our amendments.

Line 280 – Wording here is awkward – please revise Changed to clarify: "Inorganic cobalt can limit the growth of sulfate-reducing bacteria, including known Hg methylators (Ekstrom and Morel, 2008). Although inorganic cobalt (Co) serves as the center of methylcobalamin, the co-factor implied in hgcAB-mediated Hg methylation (Parks et al., 2013), Co is not known to influence Hg methylation. The significant increases in methylation from the additions of succinate and Co to oxygen minimum waters at 12 °S (Fig. 2) may indicate either a direct role of Co or a role of methylcobalamin in methylation. However, since the enhancement occurred in filtered waters rather than unfiltered water the enhanced methylation in the presence of succinate and Co in the presence of minimal cellular material warrants further study as a potential abiotic mechanism of Hg methylation."

Line 282 – Again, please revise wording "additive additions" ??? That doesn't sound right. Changed to "multiple amendments."

Line 292 - Isn't C succinate? If so I would just provide that name and remind the reader that it was a generic source of carbon. Changed.

Line 296 - What exactly do you mean by "dynamic methylation" – it sounds scientific but it delivers no meaning. Please consider changing it. You can simply describe the pattern of how and when things changed, just the way you did it in the second sentence. The use of dynamic was critiqued by both referees. As a result we have omitted its use in our revised manuscript. However, since we observed rapid methylation, rapid

demethylation, and rapid shifts between methylation and demethylation, we continue to focus our discussion on the rapid turnover between Hg species observed in our incubations.

Line 317 – But the release of Hg(II) during the two processes i.e. sinking and remineralization are connected because enhanced microbial processes are associated with sinking particles that when nearer to surface are more organic matter rich. Indeed, sinking and remineralization of particulate matter are related. However, the distinction we aimed to make here is between the importance of "residence time" of Hg(II) in the water column that appeared to promote methylation in the Heimbürger et al, 2011 study of oligotrophic waters and the delivery of particulate matter to low oxygen depths that has been observed by Sunderland et al, 2009 and Cossa et al, 2009 hypotheses. The assumption that remineralization in low oxygen water

Lines 330 - till conclusions – great discussion on the shortcomings of spiking. Ligands need to be addressed in future research. Kept in revised manuscript.

Line – 360 – this is the first I read about any effort to identify genes from Metzyme- this comes out of nowhere. The whole issue of genes here is completely unexpected – I don't see how this fits as a conclusion to this particular paper. I recommend rethinking the conclusion. We were not involved with the gene identification. These "hgcA-like" genes were observed by Podar et al, 2015 from sequences of samples collected at our incubation stations. However, the identification of these genes similar to those that encode methylation, but not containing regions that contribute to methylation, is interesting in light of the fact that bacterial methylation is commonly cited as the primary mechanism of water column methylation. The enhanced methylation we observed in filtered compared to unfiltered water incubations is consistent with

Fig. 3 – I would get rid of all the lines – they blur the figure, which already contains a lot of symbols. Perhaps you can consider splitting these two panels into more small panels? - it would show patterns more easily and then you can keep lines connecting data

for specific time points. Scale on y-axis in a) is too large. I would increase resolution of the x-axis. We have adjusted the figure to more easily distinguish the methylmercury at each time point, especially in the chlorophyll maximum.

---

## Author Comment (AC2) · 23 Jul 2018

Author comment #2

Response to general comments: We authors agree with Referee #2's critique of our failure to properly emphasize the complexity of the changes in methylation and demethylation that occurred over the course of our incubations. Our desire, in fact, was to emphasize that 24 hour rate measurements fail to describe the complexity of methylation and demethylation processes that occurred during our incubations. We presented rate constants in Table 2 to show that despite the clear variability in net methylation observed (high initial methylation followed by rapid demethylation) in our incubations, calculated rate constants were not out of range of those presented previ-

ously. Our focus during our revisions of the manuscript will be to recalculate methylation and demethylation from the time points measured and focus our discussion on those. We will include the rate constants in Table 2 to demonstrate that 24 hours is not an appropriate for quantifying accurate methylation and demethylation rates in all marine waters. It was certainly not for our stations.

Both referees recognized our failure to provide sufficient details on calculations of demethylation and the presentation of those data. This was a major deficiency of our initial manuscript and thus the major focus of our revisions.

Specific comments: Title: What is supposed to be conveyed by adding "Dynamic" to the title? Is there also a "lethargic" methylation? Or are you referring to the "Dynamics" of mercury methylation : : :? Given that both referees commented on our use of "dynamic" we will omit its use. "Dynamic" was used in contrast to "static" to indicate that 202Hg(II) can be methylated, demethylated, and methylated again (based on our time course at the 12 °S station).

L 76: terminology: rather than referring to "enriched isotope spikes" use "isotope enriched spikes. It is the spike that is enriched with isotopes not the other way around. Agreed.

L89: how exactly did you calculate ambient MMHg concentrations? What is meant by "correction for the added MM198Hg spike"? Changed to: "Estimates of MMHg at the 17 °N incubation depths were made from subtracting added MM198Hg spike contributions from the MeHg concentrations measured in initial timepoint incubations bottles (Table 1)." Because of the rapid demethylation observed in many bottles, this is only a very rough estimate of the concentrations and only provides an order of magnitude estimate at best.

L112: the concept of the "punches from McLane in situ pumps" requires more explanation. This appears to be lab lingo, which his incomprehensible to me at this point, though later on the authors shed a bit more light of what this likely means. Changed

to include details that were originally provided in lines 259- : "Treatments of carbon (1 mM, as succinate), inorganic cobalt (500 pM), and filtered particulate matter collected from McLane in situ pumps (Munson et al, 2015) were added to triplicate bottles. Pump filters were subsampled using a 2 cm (ID) acid-cleaned polycarbonate tube with a beveled edge. The 2 cm subsamples were cut in half using ceramic scissors and one of these halves was added to each sample bottle for particulate amendments."

L116: given that incubations were not performed at in-situ T, I am missing a discussion how this might have affected the outcome, since T changes alone could alter bacterial activity, leading to changes in steady state MMHg levels. The incubation temperatures were generally within a few degrees of the in situ temperature, except for the 12 °S station where we were unable to maintain the refrigerator "incubators" at sufficiently high temperatures. We will add discussion of this increase in temperature emphasizing that station and the deviations from in situ temperature as well as general temperature trends.

L145: it is an interesting concept to determine Hg(II) through direct ethylation. However, to be convinced that this is actually a viable method, I would require more QA/QC data, especially ethylation blanks. I would assume that reagents used in the methods carry some inorganic Hg background (buffers, acids, the ethylation reagent : : :) We have added the ethylation blank values in the revised manuscript. In general, the reagents carried a low but quantified THg blank that was primary due to the MQ water rather than the salts used. The ethylating reagent THg blank was below our detection limit.

L150: How are you determining a first order decay constant from at best two data points? I assume that the two points (t(0) and t(24)) themselves carry considerable uncertainty. Given than the exponential relationship, this should translate in rather large uncertainty of the resulting linear relationship and rate constant. Even if this calculation was doable (which I somehow doubt), at the very least, you should provide an uncertainty estimate, which his suspiciously absent for Kd values of table 2, while

on the other hand uncertainties for Km are provided. Equally concerning, I can't find a single data point (in a table or on a graph) for measured MM198Hg levels before, during or after incubation. This needs to be provided in order to ascertain the conclusions drawn in this paper. The error estimates for Kd values are indeed large and were omitted as an oversight. As discussed in our response to general comments, we have provided the demethylation data and methods in support of our discussion.

L158: as mentioned earlier, I think it is misleading to claim that this experiment determined "rates". Instead, it determined the net methylation that occurred over a 24 hour incubation period. In the absence of a time series showing a continuous change in concentration over time, I like challenge the idea that this dataset allows the calculation of rates, let alone rate constants. Here, the "rate" is obtained by drawing a straight line between two arbitrary points on the time axis. If the authors had chosen to incubate all samples for 6 hours, we would be facing very different "rates". This is an important point. We believe this is one of the most important findings from our data set. However since Referee #2 felt it necessary to point this out, we acknowledge our need to emphasize this point more clearly. As mentioned in our response to the general comments above, we believe it is important to present the 24 hour "rate" measurements in Table 2 to show that "reasonable," and misleading, values comparable to other published studies can be easily calculated from 24 hour data.

L210: are you sure that MMHg was indeed demethylated prior to t(0)? Have you considered other loss mechanisms, e.g. adsorption to container walls? Did you try to determine the T198Hg concentration in these samples? If there was demethylation, leading to 198Hg(II), it should show up during a total Hg determination or in the diethylHg peak of the chromatogram. If absent, what does this say about the demethylation mechanism? Would that mean the product of the demethylation is 198Hg(0)? Is that possible? Where did the 198Hg isotopes go, if they are no longer detectable as MM198Hg? Analysis of refilled sample bottles suggested little loss of MM198Hg to the bottle walls (< 1 %). As the entire sample volume was used to perform the MeHg

analysis, we could not analyze T198Hg from the same incubation bottles. Instead, we generally saw a strong 198Hg(0) peak. However, we did not attempt gaseous standard curves on the Tekran 2700 used for the analysis, so the 198Hg(0) peaks were not quantified.

L218. Be careful to not confuse "rates" with "rate constants". Demethylation rates may be expressed as the % loss per day, but this is a rather unusual expression for a rate constant, which for first order processes, has the unit of d-1 (per day). Why do you add "%" at this point? In this instance, the use of "constants" is a typo carried over from a previous version of the paragraph. Thus the units (% d-1) is correct and refers to rates and not rate constants.

L223: I agree with this concept, but I disagree in that the data obtained here are indeed "rate constants" instead, they are more net conversions over 24 h of incubation. Agreed. As discussed in our response to the general comments above, we do believe that Table 2 has its place in our manuscript since it shows that 24 h "rate constants" may mask the true rates of methylation and/or demethylation. However, our initial over interpretation of our 24 hour rate constants is unwarranted.

L243: given the absence of any actual data on MM198Hg concentrations, it is difficult to validate this conclusion. We understand. As noted, we have provided the missing MM198Hg methods and results to support our discussion of demethylation.

L259: this description of the "punch" should go to the methods section. We changed our description of the punches in the material and methods, as described in our response to L112 above.

L285; this is an intriguing observation. I'd be curious if this an experimental artifact and artificial or if this indeed points to environmental relevance for the methylation process. Certainly worth exploring in more detail. Agreed.

L294+296: "appears to be a dynamic process" what is "dynamic" in this process?

Seems to be an unnecessary filler. Please, omit "dynamic". We have omitted the use of "dynamic" as discussed in our response to the comments on the manuscript title.

L295: how did you calculate the rate of MM202Hg demethylation? Please, explain. Here the time course at 12 °S does in most cases warrant the use of ln(198MMHg) vs time and was used for the presented value. This will be clarified in the revision.

L297: I completely agree that the 24 hour incubations don't offer the resolution which would allow rate estimates with any certainty. This is not a critique of the experiment, but merely an observation. As mentioned earlier, these experiments require an educated guess about appropriate incubation periods and one only discovers after the fact, how good the initial guess really was. But rather than risking an overinterpretation of the data, the authors should rephrase their conclusions accordingly. Take into account the inherent limitations of this type of study. As noted in our response to general comments, we have restructured the manuscript to incorporate the use of methylation and demethylation (as % of spike) over incubation time periods rather than emphasizing a 24 hour rate.

L343: this observation is indeed puzzling. Can it have something to do with the acidification that is used to stop incubations? Acidification is a potential hypothesis for the rapid demethylation but is outside the scope of our study to determine. Acidification with H2SO4 is stable for MMHg and MeHg determination from environmental matrices with varying DOM concentrations as well as standards, which suggests that the rapid loss of MeHg from our incubation samples (but not the pre-equilibrated spike) may warrant additional study.

L352: another unnecessary filler: "active" seems the wrong word here, unless there is also a "passive" methylation process. We have omitted this redundancy.

L369-366: where is this discussion coming from? I fail to connect the body of this research to hgc genes. As both referees suggested that this discussion was out of
place, we will omit it from the revision. This observation was included because our measurements of enhanced methylation in filtered water are consistent with a lack of bacterial methylation as indicated from the hgc gene screening.

L367: how do you know that cellular methylation is not important, when you only determined net 24 hour methylation, rather than studying what is going on in the first 6 hours, were cellular processes may very well be important. But after 6 hours cells die (for whatever reason) and only appear to be unimportant (in the artificial setting of a closed 250 mL bottle). While this could be the case of unfiltered water incubations, in most cases we observed no enhanced methylation in unfiltered water relative to filtered water. The 250 mL is indeed an artificial environment, but the differences in methylation between filtered and unfiltered water, rather than the duration of methylation in unfiltered water, is the basis of our conclusion.

Figure 2: What is the difference between panel a+b and c+d? There is no legend for panels a+c. Was the concentration of the Hg(II) substrate determined (how?) or is this the nominal spike concentration? The figure captioned has been revised. The Hg(II) substrate was determined from the quantification of the diethylHg peak from the chromatogram as outlined in the methods. The identification of filtered (a+c) and unfiltered (b+d) results was not transferred when we adjusted the figure labels and is now included in the caption.

Figure 3: the chosen presentation makes it very difficult, if not impossible for most treatments to decide if concentrations after 24 hours are smaller or larger compared to the t(0) starting point. Agreed. As noted in our response to Referee #1, we have removed the lines and split the figure panels and resulting y axis scales to clarify.

Table 1: typo for THg of 17N CMX: 1:0. This is not a typo, but we will clarify. Because of water restrictions on the cast used to collect water at the 17 °N station, we had to mix water from 2 depths. Thus the presented concentrations "1:0.59" are the concentrations in the water from 120 m and 150 m. However, since this is confusing, we will

change to "0.59-1.00" for clarification.

Table2: Why are there no uncertainty estimates for Kd values? The uncertainty estimates were not transferred into the submitted manuscript version. However, the more egregious error is likely the need for a better calculation of demethylation. Our revision focuses on the % of isotope tracer demethylated and the accompanying error in our calculations.